# Autoimmune anti-DNA and anti-phosphatidylserine antibodies predict development of severe COVID-19

Claudia Gomes[1,*], Marisol Zuniga[1,*], Kelly A Crotty[1] , Kun Qian[2], Nubia Catalina Tovar[1,3,4], Lawrence Hsu Lin[5], Kimon V Argyropoulos[5], Robert Clancy[6], Peter Izmirly[6], Jill Buyon[6], David C Lee[7], Maria Fernanda Yasnot-Acosta[3] , Huilin Li[2], Paolo Cotzia[5], Ana Rodriguez[1] 

**High levels of autoimmune antibodies are observed in COVID-19 patients but their specific contribution to disease severity and clinical manifestations remains poorly understood. We performed a retrospective study of 115 COVID-19 hospitalized patients with different degrees of severity to analyze the generation of autoimmune antibodies to common antigens: a lysate of erythrocytes, the lipid phosphatidylserine (PS) and DNA. High levels of IgG autoantibodies against erythrocyte lysates were observed in a large percentage (up to 36%) of patients. Anti-DNA and anti-PS antibodies determined upon hospital admission correlated strongly with later development of severe disease, showing a positive predictive value of 85.7% and 92.8%, respectively. Patients with positive values for at least one of the two autoantibodies accounted for 24% of total severe cases. Statistical analysis identified strong correlations between anti-DNA antibodies and markers of cell injury, coagulation, neutrophil levels and erythrocyte size. Anti-DNA and anti-PS autoantibodies may play an important role in the pathogenesis of COVID-19 and could be developed as predictive biomarkers for disease severity and specific clinical manifestations.**

## Introduction

Infections trigger immune responses that target pathogen antigens, but frequently they also induce potent autoimmune responses that are characterized by high levels of antibodies recognizing a variety of host antigens (Rivera-Correa & Rodriguez, 2018). Autoimmune antibodies have been characterized in viral diseases such as hepatitis C, HIV, and arboviral infections, but also in bacterial and protozoan infections like tuberculosis, and malaria. Infection-induced autoantibodies can recognize a variety of self-antigens,

including nucleic acids, lipids, and glycoproteins (Rivera-Correa & Rodriguez, 2018).

Autoimmune antibodies can contribute to systemic inflammatory responses and subsequent tissue damage through different mechanisms, including immune complex (IC) formation, complement activation, formation of thrombi, and/or lysis of uninfected cells (Ludwig et al, 2017).

In COVID-19, autoantibodies are found in high levels in a large proportion of hospitalized patients with severe disease (Woodruff et al, 2020 Preprint; Zhou et al, 2020; Wang et al, 2021) and have been associated with the development of autoimmune pathologies (Ehrenfeld et al, 2020), such as thrombocytopenia (Zulfiqar et al, 2020), hemolytic anemia (Lazarian et al, 2020), Guillain–Barre (Toscano et al, 2020), and anti-phospholipid (Zhang et al, 2020; Zhou et al, 2020) syndromes. Autoantibodies against type I interferons (Bastard et al, 2020) or a panel of cytokines (Wang et al, 2021) have also been identified and may explain disease severity among subsets of COVID-19 patients. Autoantibodies against the lung protective protein Annexin-A2 correlated with patient mortality in COVID-19, suggesting a detrimental effect of certain autoantibody specificities in patients (Zuniga et al, 2021).

In addition to acute respiratory distress and pulmonary edema, COVID-19 can cause multi-organ widespread thrombosis and disseminated intravascular coagulation (Merrill et al, 2020). To explore the relation of autoimmune antibodies to COVID-19 clinical manifestations, we have determined IgG autoantibody levels in COVID-19 patients' plasma against a lysate of erythrocytes (as a general measure of autoreactivity) and two specific antigens that are involved in the autoimmune pathogenesis of other diseases: anti-phosphatidylserine (PS) (Rivera-Correa & Rodriguez, 2020) and anti-DNA (Pisetsky & Lipsky, 2020). Our analysis showed that some hospitalized patients present high levels of these autoantibodies and that there is a strong correlation among these levels in patients, pointing to general autoreactivity as a common phenomenon in COVID-19 patients. Among hospitalized patients, severity of disease

[1]Department of Microbiology, New York University Grossman School of Medicine, New York, NY, USA [2]Division of Biostatistics, Department of Population Health, New York University Grossman School of Medicine, New York, NY, USA [3]Universidad de Córdoba, Montería, Córdoba, Colombia [4]Universidad Del Sinú, Montería, Córdoba, Colombia [5]Department of Pathology, New York University Grossman School of Medicine, New York, NY, USA [6]Division of Rheumatology, Department of Medicine, New York University Grossman School of Medicine, New York, NY, USA [7]Department of Emergency Medicine, New York University Grossman School of Medicine, New York, NY, USA

Correspondence: ana.rodriguez@nyumc.org
*Claudia Gomes and Marisol Zuniga contributed equally to this work

was strongly correlated with levels of anti-DNA antibodies and weakly with anti-PS. High-throughput data analysis showed that anti-DNA antibodies correlate strongly with parameters related to cell injury, coagulation, neutrophil levels and erythrocyte distribution width, suggesting a role of these autoantibodies in exacerbating COVID-19 clinical course of disease.

# Results

## Patients characteristics

115 cases of hospitalized patients with SARS-CoV-2 infection with different degrees of disease severity and 20 uninfected controls were used in this retrospective study. Patients were stratified into three different groups: nonsevere, severe patients that survived, and severe patients that died of COVID-19 (Table 1).

## Autoantibody levels are elevated in hospitalized patients with COVID-19 compared with controls

The plasma of controls and COVID-19 patients on day 0–1 was used to determine the levels of different IgG autoimmune antibodies. As a general measurement of the autoimmune antibody response, samples were tested for reactivity to a lysate of human erythrocytes (RBCL). Autoantibodies to the phospholipid PS and DNA were also determined because they have been involved in the pathogenesis of other diseases (Pisetsky & Lipsky, 2020; Rivera-Correa & Rodriguez, 2020).

We observed that COVID-19 patients had significantly higher average levels of circulating anti-RBCL, anti-PS, and anti-DNA than controls, with 36% of patients testing positive for anti-RBCL (Fig 1A–C). Patients with systemic lupus erythematosus (SLE), an autoimmune disorder which frequently results in increased levels of anti-DNA antibodies (Pisetsky, 2016), showed higher levels of all autoantibodies, especially of anti-DNA than COVID-19 patients. Patients with an infectious disease that also induces high levels of different autoantibodies, malaria (Rivera-Correa & Rodriguez, 2020), showed higher levels of anti-PS and anti-DNA, but lower levels of anti-RBCL (Fig 1A–C).

**Table 1. Demographic and clinical characteristics of patients (n = 115).**

| | |
|---|---|
| Age yr, median (IQR) | 66 (20–101) |
| Female sex, n (%) | 35 (30.4) |
| Race, n (%) | |
| White | 68 (59.1) |
| Black | 14 (12.2) |
| Asian | 8 (7.0) |
| Other | 25 (21.7) |
| Disease severity, n (%) | |
| Non-severe | 40 (34.8) |
| Severe that survived | 42 (36.5) |
| Severe that died | 33 (28.7) |

The specificity of anti-DNA antibodies was similar in all three diseases, COVID-19, SLE and malaria, with broad reactivity to single stranded (ss)DNA, double stranded (ds)DNA, and CpG DNA (Fig 2A–D).

In COVID-19 patients, the levels of autoantibodies increased with severity of disease, showing higher levels in severe patients than in nonsevere or control (Fig 3A–C).

In patients with autoimmune diseases, binding of autoantibodies to their antigens results in the formation of ICs, which are deposited in various tissues frequently leading to disease (Ludwig et al, 2017). The levels of circulating IC were determined in this cohort, finding no increases between controls and COVID-19 patients (Fig 4).

The levels of the three different autoantibodies that we examined are highly correlated with each other (Table 2), indicating that individual patients tend to have similar levels of different autoantibodies and suggesting some patients are more prone to autoreactivity.

When the severity of disease or death occurrence were analyzed in regard to the levels of autoantibodies, we observed a strong correlation of anti-DNA antibody levels (OR = 7.2, $P$ = 0.006) and a moderate correlation of anti-PS antibodies (OR = 5.7, $P$ = 0.043) with disease severity after adjustment for age, race, and sex (Table 3).

We observed that a large proportion of patients that were positive for anti-DNA or anti-PS antibodies upon hospital admission developed severe disease. Anti-DNA antibodies showed a positive predictive value of 86% for COVID-19 severity (Table 4), which corresponds to 8% of total severe cases in this cohort (6 of 75), whereas anti-PS antibodies showed a positive predictive value of 93%, corresponding to 17% of total severe cases (13 out 75). Patients with positive values for at least one of the two autoantibodies correspond to 24% of total severe cases (18 of 75). Only one of the 18 patients was positive for both autoantibodies.

## Relationship between autoantibody levels and clinical parameters

We first performed a statistical analysis comparing the levels of the three determined autoimmune IgG antibodies and IC with values of laboratory tests assessed in the 115 patients hospitalized in the cohort. These include data from 118 parameters that were assessed in the context of hematologic, metabolic, immunological, and biochemical monitoring of these patients during their admission.

No significant correlations were found between the levels of any of the autoantibodies or IC with any of the clinical parameters measured at day 0–3 (Table S1). However, significant correlations ($\rho$ > 0.3 and $P$ < 0.01) were found between the levels of different autoantibodies and the maximum value of several laboratory tests performed during each patient stay at the hospital. The correlations of each autoantibody type were distinct from the others and tended to be clustered in groups of parameters related to specific disease processes (Fig 5). Analysis of the minimum values of laboratory tests did not identify any significant correlations with autoantibody levels (Table S2).

Anti-DNA antibodies correlated strongly with the maximum values of two markers of cellular injury: lactate dehydrogenase (LDH), which is released by all cell types, and creatine kinase, which is released specifically by striated muscle cells, suggesting a

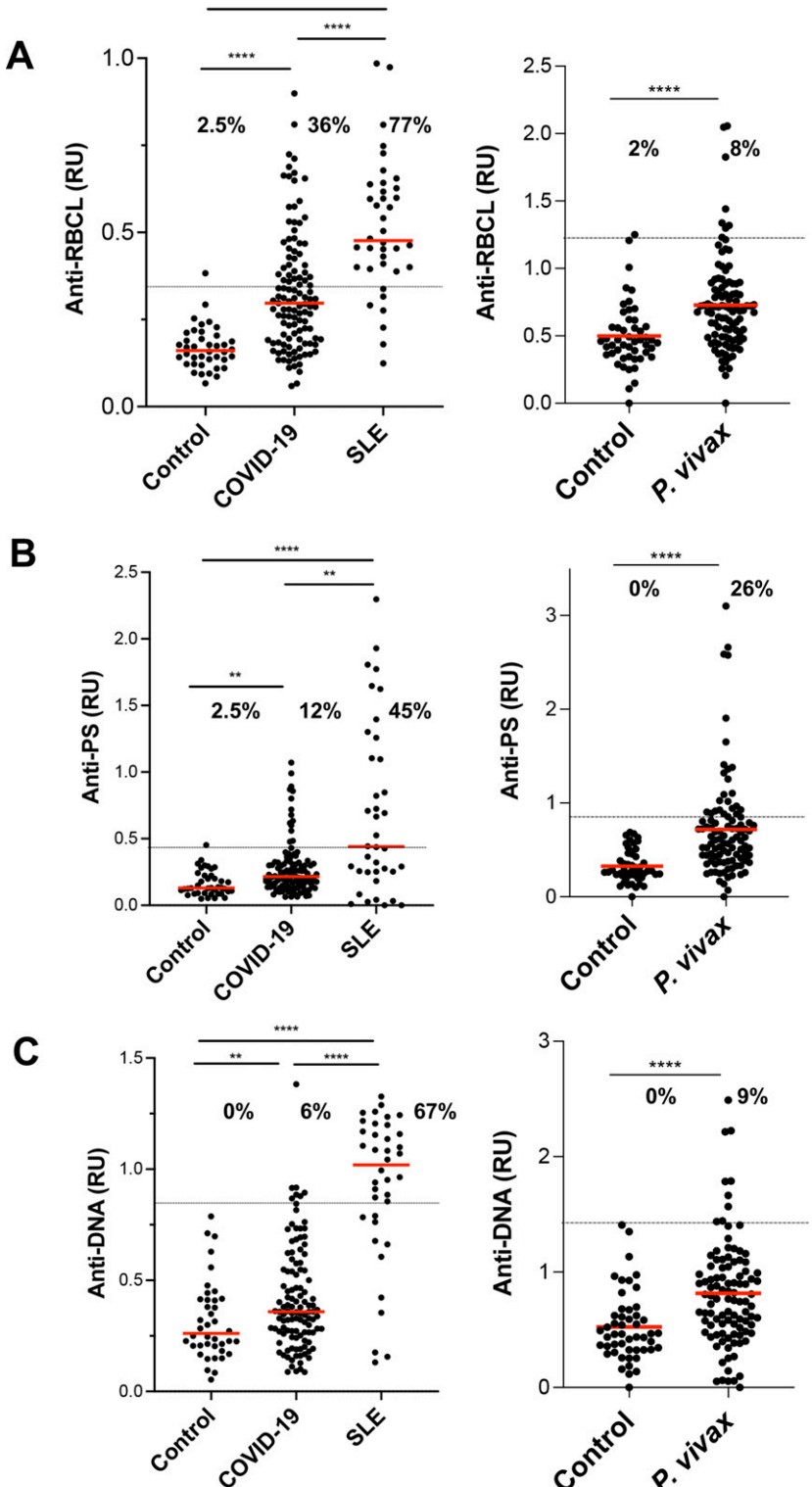

**Figure 1. COVID-19 patients present higher levels of IgG autoimmune antibodies than uninfected controls.**
**(A, B, C)** Analysis of plasma samples from 115 COVID-19, 38 systemic lupus erythematosus (SLE), and 101 *Plasmodium vivax* malaria patients for levels of IgG, anti-RBCL (A), anti-PS (B), and anti-DNA (C) by ELISA. Control samples were collected in the same area as patient samples, New York City for COVID-19 and SLE, and Tierralta (Colombia) for *P. vivax* malaria. Samples were considered positive for autoantibodies if the relative units > the mean plus three times the standard deviation of the controls. Percentage of positive samples for each group is indicated. The cut-off is indicated by the dashed horizontal line. Average of duplicated values for each sample is shown. *P < 0.05, **P < 0.001, ***P < 0.0001, ****P < 0.0001, Man–Whitney test.
Source data are available for this figure.

possible link between anti-DNA antibodies and cellular lysis (Fig 5). Significant correlations were also found between anti-DNA antibodies and absolute numbers of neutrophils, total white blood cell counts and markers of RBCs volume, such as mean corpuscular volume and red cell distribution width (RDW). We also observed that maximum values of D-dimer concentration, a biomarker for coagulation disorders and thrombus formation, correlated significantly with the levels of anti-DNA antibodies.

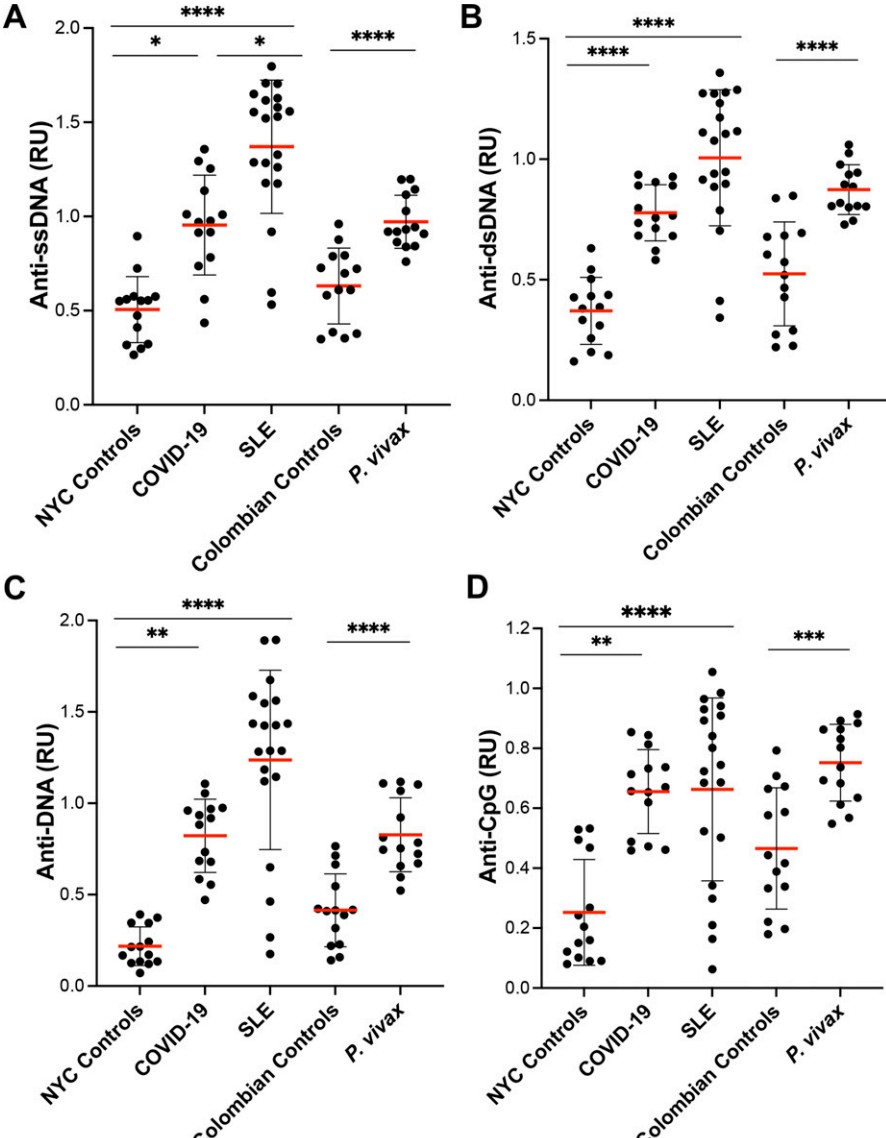

**Figure 2. Anti-DNA antibodies from COVID-19 patients recognize ssDNA, dsDNA, and CpG.**
**(A, B, C D)** Plasma samples from COVID-19 (n = 14) and systemic lupus erythematosus (n = 20) patients and controls from New York City (n = 14), and *Plasmodium vivax* malaria patients (n = 14) and controls from Tierralta (Colombia) (n = 14) were tested for levels of IgG to ssDNA (A), dsDNA (B), DNA as used in Fig 1 (C), and CpG (D) by ELISA. Average of duplicated values for each sample with standard deviation is shown. ***P < 0.0001, ****P < 0.0001, Mann–Whitney test.
Source data are available for this figure.

Anti-PS antibodies correlated positively with platelet levels. Anti-RBCL correlated with the total levels of protein in plasma and parameters related to RBC. IC levels correlated with parameters related to kidney function, as it is typical in autoimmune disorders.

It is important to note that no correlation was found between any of the autoantibodies and cytokines or inflammatory mediators such as C-reactive protein, several interleukins (including IL6), TNF, or interferon-γ.

## Discussion

Although autoantibodies have been observed in different viral and nonviral infections (Rivera-Correa & Rodriguez, 2018), the high percentage of COVID-19 patients with autoantibodies (up to 36% positive for anti-RBCL) and the strong correlation of anti-RBCL

autoantibodies and total protein levels in plasma, indicates that the levels of autoreactivity are particularly high in this infection. Similarly, a study of autoantibodies to protein antigens in COVID-19 patients also found high general autoreactivity (Wang et al, 2021).

COVID-19 patients presented similar levels of anti-RBCL, anti-PS, and anti-DNA as patients with malaria, another highly inflammatory infectious disease, but lower levels than patients with SLE. Although the levels of anti-DNA antibodies in COVID-19 patients were not as high as they are observed in SLE, the specificity of these autoantibodies was similarly broad, recognizing both ssDNA, dsDNA, and CpG. One possible limitation of this study is that the determinations were performed using ELISA, which is less specific than immunofluorescence tests with *Crithidia luciliae* (Granito et al, 2021).

We observed that the levels of these three autoantibodies determined are highly correlated with each other in the three diseases studied, which is consistent with the activation of a generalized autoimmune response in certain patients. However, we

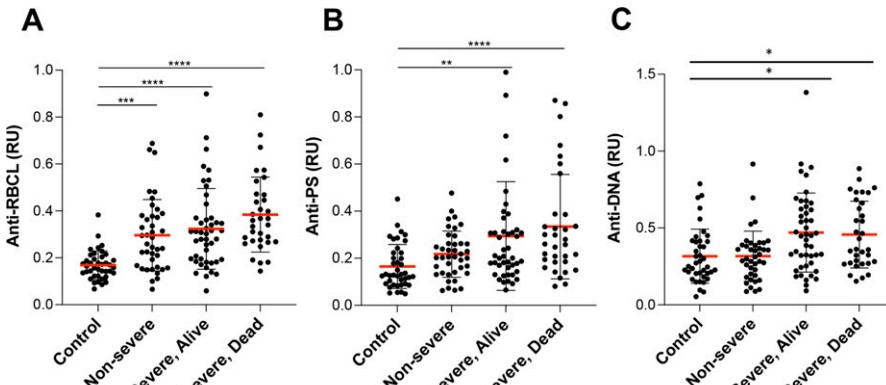

**Figure 3. Autoantibody levels increase with severity of disease.**
**(A, B, C)** Comparison of controls and COVID-19 patients stratified by severity of disease for levels of IgG, anti-RBCL (A), anti-PS (B), and anti-DNA (C). Average of duplicated values for each sample with standard deviation is shown. *P < 0.05, **P < 0.001, ***P < 0.0001, ****P < 0.0001, Mann–Whitney test.
Source data are available for this figure.

observed a higher proportion of COVID-19 patients with elevated levels of circulating anti-RBCL and anti-PS compared with anti-DNA, which may suggest that specific autoantibodies are produced more frequently in our cohort, but also that antibodies with certain specificities may be removed from the circulation more efficiently.

In contrast to other infectious diseases that also induce auto-antibodies (Rivera-Correa & Rodriguez, 2018), we observed that the levels of IC are not increased in the circulation of COVID-19 patients. This is unexpected, considering the high levels of autoimmune antibodies with specificities for antigens that are likely to be accessible in the circulation, such as PS in exosomes or cell-free DNA (Cheng et al, 2021). However, it is possible that IC are being formed, but are rapidly removed from the circulation, which would be consistent with the observed correlations between levels of IC and parameters indicating kidney disfunction.

It is important to note that general levels of autoreactivity, that is, anti-RBC autoantibodies, do not correlate with the development of severe disease, confirming that anti-DNA and anti-PS are specific indicators. Other studies have analyzed autoantibodies of different specificities, finding that protein antigens such as type I interferons (Bastard et al, 2020), particular tissue associated antigens and cytokines (Wang et al, 2021) or the lung protective protein Annexin-A2 (Zuniga et al, 2021) correlate with COVID-19 severity or death in specific subgroups of patients.

Our study is focused on two specific autoantibodies, anti-DNA and anti-PS, because of their previously described roles in the pathogenesis of other diseases (Pisetsky & Lipsky, 2020; Rivera-Correa & Rodriguez, 2020) and their capacity to bind to different cell types in the circulation (Fernandez-Arias et al, 2016; Tamkovich & Laktionov, 2019). We observed that anti-DNA antibodies, which are typically found in autoimmune disorders, such as SLE (Pisetsky, 2016) and in some infectious diseases (Berlin et al, 2007), are found in 6% of COVID-19 patients in this cohort and correlate strongly with severity of COVID-19 disease in this group. Similarly, anti-PS antibodies, which contribute to anemia in malaria patients (Fernandez-Arias et al, 2016), are found in 12% of COVID-19 patients and also correlate with severity of disease. Because the levels of anti-DNA and anti-PS were determined upon hospital admission, it is possible that these simple tests could be developed as a predictor of COVID-19 severity. In this cohort, high anti-DNA and anti-PS antibodies accounted for 24% of total severe patients.

COVID-19 patients have high levels of cell-free DNA in the circulation, which are also related to disease severity (Lee et al, 2020; Cheng et al, 2021). Furthermore, we observed a strong correlation between markers of cell injury and anti-DNA levels, which mirrors the previously found correlation between these markers and cell-free DNA levels (Cheng et al, 2021). Because the levels of cell-free DNA and anti-DNA correlate with severity of disease and markers of cell injury, it is likely that the binding of anti-DNA to cell-free DNA contributes to the pathogenesis of COVID-19 manifestations.

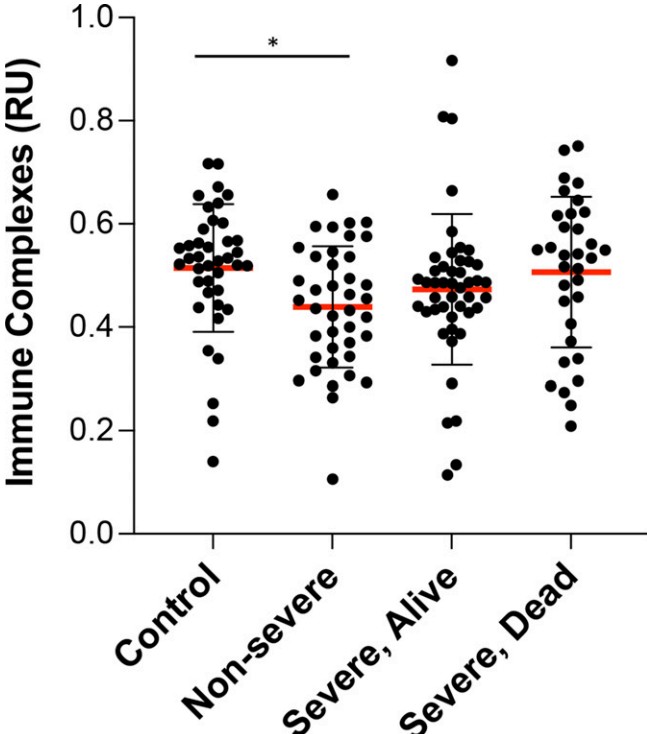

**Figure 4. COVID-19 patients do not present higher levels of immune complex.**
Analysis of 115 plasma samples of COVID-19 patients stratified by severity of disease and 40 controls for levels of C1q-binding immune complex by ELISA. Average of duplicated values for each sample with standard deviation are shown. *P < 0.05, Mann–Whitney test.
Source data are available for this figure.

**Table 2. Correlation[a] among autoantibody and immune complex levels.**

| COVID-19 (n = 115) | Anti-PS | | Anti-DNA | | Immune complex | |
|---|---|---|---|---|---|---|
| | ρ | P-value | ρ | P-value | ρ | P-value |
| Anti-RBCL | **0.61** | **<0.0001** | **0.50** | **<0.0001** | **0.41** | **<0.0001** |
| Anti-PS | - | - | **0.46** | **<0.0001** | **0.36** | **<0.0001** |
| Anti-DNA | - | - | - | - | 0.23 | 0.0151 |
| Systemic lupus erythematosus (n = 40) | | | | | | |
| Anti-RBCL | 0.37 | 0.020 | **0.52** | **0.0008** | | |
| Anti-PS | - | - | **0.43** | **0.0070** | | |
| Malaria (n = 101) | | | | | | |
| Anti-RBCL | **0.46** | **<0.0001** | **0.37** | **0.0001** | | |
| Anti-PS | - | - | **0.54** | **<0.0001** | | |

Bold indicates correlations with $p > 0.3$ and $P < 0.01$.
[a]Spearman correlation test.

It is known that cell-free DNA binds to the surface of endothelial and immune cells (Tamkovich & Laktionov, 2019), as well as erythrocytes (Hotz et al, 2018), where it could constitute a target for anti-DNA antibodies in the circulation, triggering complement-mediated cell lysis. We observed a strong correlation of anti-DNA antibodies with lactate dehydrogenase (LDH) and creatine kinase, suggesting that anti-DNA antibodies may contribute to muscle injury, which is frequent in COVID-19 patients (Paliwal et al, 2020).

High levels of cell-free DNA in COVID-19 patients are probably a result of the active release of Neutrophil Extracellular Traps (NETs) which are mainly composed of neutrophil DNA (Middleton et al, 2020). Interestingly, anti-DNA antibodies also correlate with absolute levels of neutrophils.

Anti-DNA antibodies also correlate with a marker of thrombosis, D-dimer, but not with other parameters involved in triggering the coagulation cascade. Because NETs contribute to thrombosis and have been found to be components of the micro-thrombi in COVID-19 patients (Middleton et al, 2020), it is possible that anti-DNA antibodies bind to the DNA in NETs, facilitating cellular aggregation and possibly contributing to intravascular coagulation.

Anti-DNA antibodies also correlate with a parameter that determines the variation of erythrocyte volume, RDW, which has been associated with mortality risk in multiple diseases, including COVID-19 (Foy et al, 2020).

Anti-PS antibodies correlated with high levels of platelets. Although a low platelet count is associated with increased risk of severe disease and mortality in patients with COVID-19 (Lippi et al, 2020), it has also been observed that platelets are hyperactivated during the disease (Zaid et al, 2020) and that immature platelets (Welder et al, 2021) and coated-platelet levels (Khattab et al, 2021) correlate with severity and death in COVID-19 patients, suggesting that they may be involved in thrombo-inflammation and contribute to disease severity.

Severe complications in COVID-19 patients typically develop at least 1 wk after the onset of symptoms (Polak et al, 2020), when viral levels are already decreasing and are frequently undetectable (Sethuraman et al, 2020). These observations suggest that the most severe forms of disease observed in COVID-19 patients may be a result of the host response to infection, rather than a direct consequence of viral cytopathic effect. Autoantibodies, as part of the host response to infection, may contribute to this delayed pathogenesis through different mechanisms. Our work suggests that anti-DNA and anti-PS antibodies may have a role in different pathogenic processes, including cell injury and coagulation, constituting a possible mechanism contributing to pathogenesis in COVID-19 patients.

# Materials and Methods

### Bioethics statement

The collection of COVID-19 human bio-specimens for research has been approved by New York University Langone Health (NYULH) Institutional Review Board under the S16–00122 Universal Mechanism of human bio-specimen collection and storage for research. This protocol allows the collection and analysis of clinical and demographic data. The database used for this project is de-identified.

**Table 3. Correlation analysis between autoantibodies and severity/living status of hospitalized patients.**

| | Odds ratio[a] | P-value[b] |
|---|---|---|
| Living status[c] | | |
| Anti-RBCL | 3.921 | 0.137 |
| Anti-PS | 2.083 | 0.161 |
| Anti-DNA | 1.925 | 0.161 |
| Immune complex | 4.759 | 0.137 |
| Severity[c] | | |
| Anti-RBCL | 2.741 | 0.119 |
| Anti-PS | **5.765** | **0.043** |
| Anti-DNA | **7.219** | **0.006** |
| Immune complex | 3.477 | 0.119 |

Bold indicates Odds ratio > 5 and $P < 0.05$.
[a]1 U = 0.5 relative unit.
[b]False discovery rate adjusted P.
[c]Regression analysis adjusted for age, sex, and race.

**Table 4. Autoantibodies positive predictive value for disease severity in hospitalized patients.**

| | Positive predictive value, % (positives/total)[a] |
|---|---|
| Anti-RBCL | 66.6 (28/42) |
| Anti-PS | 92.8 (13/14) |
| Anti-DNA | 85.7 (6/7) |
| Immune complex | 0 (0/0) |

[a]Patients that develop severe disease from total autoantibody or immune complex positive at day 0.

## Study design and participants

This retrospective cohort study included 115 hospitalized cases of SARS-CoV-2 infection at NYULH and 42 uninfected controls. All COVID-19 cases occurred from April until June 2020, during the peak of the COVID-19 pandemic in this area. Control samples used were collected earlier than January 2020, to avoid possible undiagnosed COVID-19 cases. Patients that needed intubation or required high flow oxygenation and intensive care monitoring were considered severe.

## Sample collection and testing for COVID-19

Sample collection was performed as part of routine clinical care for COVID-19 patients at NYULH at the time of hospital admission (day 0 or 1). Venous blood samples were collected in standard plasma separation tubes containing heparin (16 USP U/ml) (BD Diagnostics). Plasma was recovered after centrifugation, stored at 5°C for 5 d (in case further clinical testing for the patient was required) before aliquoting and storage at −80°C. Assessment of SARS-CoV-2 infection was performed using nasopharyngeal swabs in clinical PCR assays as described (Maurano et al, 2020).

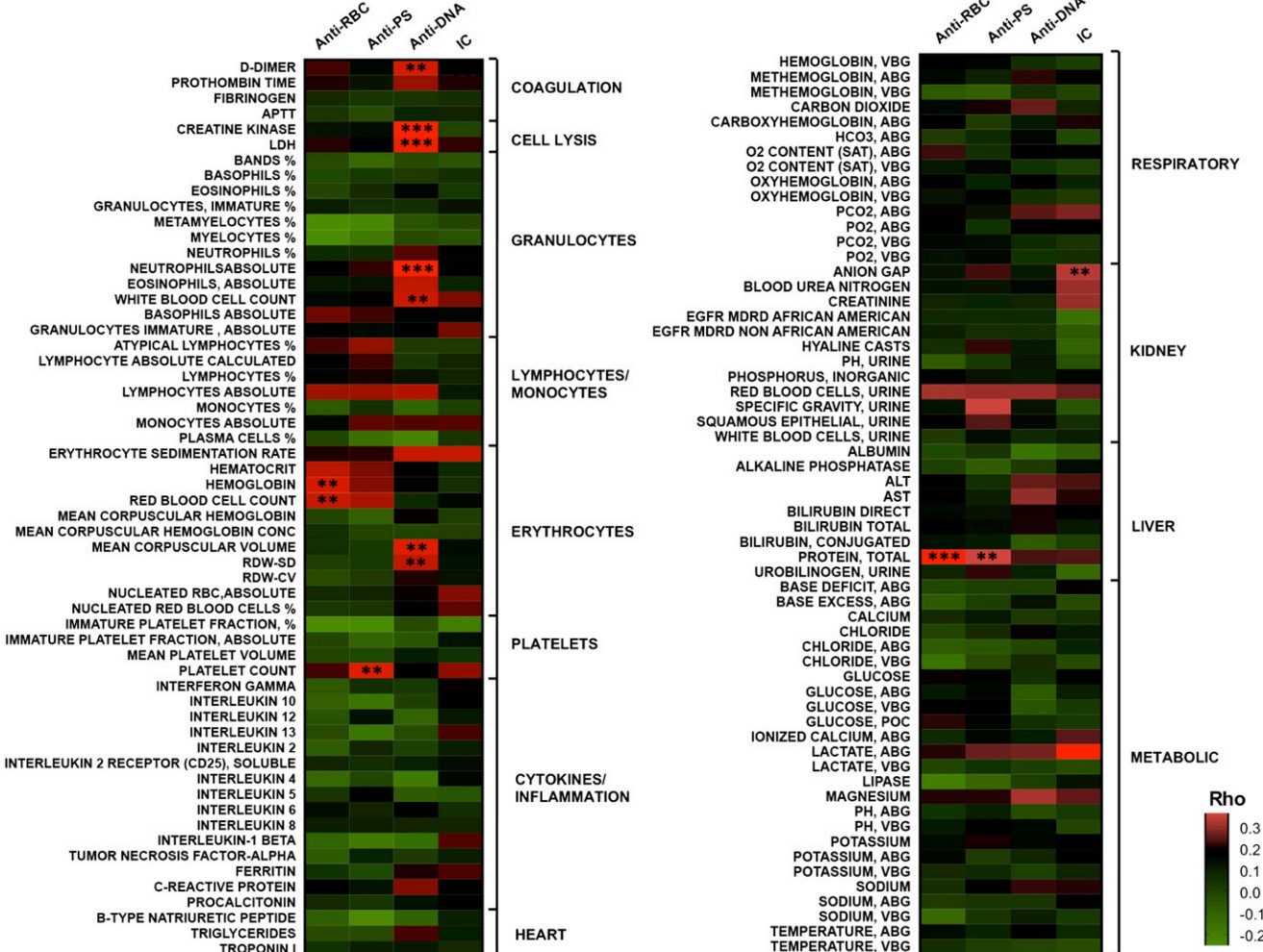

**Figure 5. Correlation analysis of autoantibodies and maximum values of clinical tests identified significant associations mainly for anti-DNA antibodies.**
Heat map listing 118 clinical tests correlation with the autoantibodies and immune complex. Color scale indicates $\rho$ values for each pair in the two tailed Spearman correlation test. Corresponding P-value (with false discovery rate correction) is indicated if $P < 0.001$ (**$P < 0.001$, ***$P < 0.0001$).
Source data are available for this figure.

## Sample collection and testing for SLE

Sera were obtained from patients previously recruited to the New York University (NYU) Lupus Cohort, a prospective convenience registry open to enrollment of any patient with SLE seen at NYU Langone Health and Bellevue Hospital Center since 2014. All SLE patients in the NYU Lupus Cohort are age 18 or older and fulfill sufficient criteria to be classified as SLE by at least one of the following: (1) the American College of Rheumatology revised classification criteria (Hochberg, 1997); (2) the Systemic Lupus International Collaborating Clinics classification criteria (Petri et al, 2012); (3) the European League Against Rheumatism (EULAR)/American College of Rheumatology classification criteria (Aringer et al, 2019). All NYU lupus cohort patients signed an informed consent approved the NYU and Bellevue Hospital Institutional Review Boards, which is available in English, Spanish, and Mandarin. For the current study, patients were selected based on previously recorded commercial values for the presence of antibodies to dsDNA (assayed by BioPlex and ELISA) with the intent to evaluate a spectrum of titers. A comparison of anti-dsDNA titer for the 20 SLE samples as determined by BioPlex and to calf thymus DNA by ELISA, as described below, showed a Spearman correlation with r = 0.86 and P < 0.0001, indicating that both methods provide similar relative quantifications of anti-DNA antibodies in the patient samples.

## Sample collection and testing for *Plasmodium vivax* malaria

Plasma samples from complicated and uncomplicated *P. vivax* infections and controls were collected from patients recruited at Hospital San Jerónimo of Monteria and Hospital San José of Tierralta, Córdoba, Colombia, between 2017 and 2019 as described before (Rivera-Correa et al, 2020).

## Clinical parameters analysis

Every patient sample is associated with a list of parameters, including non-identifiable information (age, sex, and race), clinical data and complete immunological, hematological, and biochemical clinical characterization. The clinical parameters used for the analysis had data from more than 30 patients (n > 30). The analysis was performed to determine statistical correlations between the levels of autoantibodies, which were determined on day 0–1 of admission and clinical parameters determined on day 0–3, to include some tests that were performed a few days after admission. A parallel analysis was also performed to analyze statistical significance in the relation between autoantibody levels and the minimum or maximum value obtained for every laboratory test during each patient stay at the hospital, with the aim to determine the predictive value of autoantibody levels for the different clinical parameters.

## Data analysis

SQL statements were generated programmatically to select specific records from NYU de-identified COVID-19 database. WHERE clauses were generated so that only specific patient records were selected. Records of interest were then exported to Excel using the HUE SQL query tool. Excel worksheets were generated using the Infragistics

Excel Engine software library (Infragistics, Inc.) which synchronized the data records obtained from the NYU COVID-19 database with de-identified numbers which correspond to patient samples. A simple algorithm was applied to the laboratory result records to obtain the minimum and maximum values for each laboratory test and create new worksheets, designed to be consumed by statistical analysis software.

## Statistical analysis

Data were analyzed using GraphPad Prism v8. Two-tailed Spearman correlation was used to evaluate the association between the different autoantibodies and between the autoantibodies and clinical parameters. False discovery rate was controlled for multiple comparisons within specific groups. The differences in autoantibody levels between controls and COVID-19 patients were determined using Mann–Whitney test. Because the levels of autoantibodies do not follow a Gaussian distribution, the confidence levels of the average plus three times the standard deviation is 89% calculated by Chevyshev bound. The association between living status and severity of disease with the autoantibodies was performed using logistic regression, adjusting for age, race, and gender.

## ELISA

To measure autoimmune antibodies, Immulon 2HB 96-well ELISA plates were coated with lysates of human erythrocytes ($2 \times 10^3$ cells/$\mu$l in PBS, from Interstate Blood Bank), PS (Sigma-Aldrich) at 20 $\mu$g/ml in 200-proof Molecular Biology ethanol, calf thymus DNA (predominantly double stranded, D4522; Sigma-Aldrich), purified ssDNA (D8273; Sigma-Aldrich), dsDNA (D8515; Sigma-Aldrich) or CpG (at 10 $\mu$g/ml in PBS). Plates were allowed to evaporate for 16 h at 4°C. PS-coated plates were let evaporate at RT until completely dry. Plates were washed three times with PBS 0.05% Tween-20 and then blocked for 1 h at 37°C with PBS 3% BSA. Plasma from patients or control was diluted at 1:100 in blocking buffer and incubated in duplicates for 2 h at 37°C. Plates were washed again three times and incubated with a polyclonal goat anti-human IgG-HRP diluted 1:500 (Invitrogen) for 1 h at 37°C. Plates were washed four more times and developed using TMB substrate (BD Biosciences). The reaction was stopped using Stop buffer (BioLegend) and absorbance read at 450 nm. The mean OD at 450 nm from duplicate wells was compared with a reference positive control plasma sample, previously identified as positive for IgG autoantibodies and IC, to calculate relative units. To determine IC, plates were coated with C1q polyclonal antibody (1 $\mu$g/ml in carbonate-bicarbonate buffer; Invitrogen) and the same protocol was followed with the following modifications: plates were blocked for 1 h 30 min with PBS 0.1% BSA, plasma and secondary antibody were diluted 1:80 and 1:5,000, respectively. All steps were performed at room temperature. Samples were considered positive for autoantibodies if the relative units were greater than the mean plus three times the standard deviation of the controls.

## Role of the funding source

The funder had no role in study design, data collection, data analysis, data interpretation, or writing of the report. The corresponding

author had full access to all data in the study and had final responsibility for the decision to submit for publication.

## Data Availability

This study includes no data deposited in external repositories.

## Supplementary Information

## Acknowledgements

We thank the personnel at the New York University (NYU) School of Medicine Center for Biospecimen Research and Development and Brian Fallon for bioinformatics support. This research was supported by the NYU Langone COVID-19 special fund, National Institutes of Health/National Institute of Arthritis and Musculoskeletal and Skin Diseases P50 AR07059 and Bloomberg Philanthropies COVID-19 Response Initiative Grant.

### Author Contributions

C Gomes: data curation, investigation, and methodology.
M Zuniga: data curation, investigation, and methodology.
KA Crotty: data curation, investigation, and methodology.
K Qian: formal analysis.
NC Tovar: data curation, investigation, and methodology.
L Hsu Lin: resources, data curation, investigation, methodology, and patient recruitment and sample collection.
KV Argyropoulos: resources, patient recruitment and sample collection, and writing—review and editing.
R Clancy: resources.
P Izmirly: resources.
J Buyon: resources.
DC Lee: conceptualization and formal analysis.
MF Yasnot-Acosta: resources.
H Li: formal analysis.
P Cotzia: patient recruitment and sample collection and writing—review and editing.
A Rodriguez: conceptualization, data curation, formal analysis, supervision, funding acquisition, and writing—original draft, review, and editing.

### Conflict of Interest Statement

The authors declare that they have no conflict of interest.

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
