## [Reviewer comments · Life Science Alliance]

Life Science Alliance

Autoimmune anti-DNA and anti-Phosphatidylserine antibodies predict development of severe COVID-19

Claudia Gomes, Marisol Zuniga, Kelly Crotty, Kun Qian, Nubia Catalina Tovar, Lawrence Hsu Lin, Kimon Argyropoulos, Robert Clancy, Peter Izmirly, Jill Buyon, David Lee, Maria Fernanda Yasnot-Acosta, Huilin Li, Paolo Cotzia, and Ana Rodriguez

DOI: <https://doi.org/10.26508/lsa.202101180>

Corresponding author(s): Ana Rodriguez, New York University School of Medicine

Review Timeline:

Submission Date:	2021-07-29
Editorial Decision:	2021-08-02
Revision Received:	2021-08-04
Editorial Decision:	2021-08-06
Revision Received:	2021-08-18
Accepted:	2021-08-18

Transaction Report:

Please note that the manuscript was previously reviewed at another journal and the reports were taken into account in the decision-making process at Life Science Alliance.

1st Review Round

Referee #1 Review

Remarks for Author:

Several reports have documented high levels of autoantibodies in hospitalized COVID-19 patients with severe disease, and it is now believed that autoantibodies play an important role in COVID-19 pathogenesis. In this study, Gomes et al. report the levels of anti-RBCL, anti-PS, and anti-DNA autoantibodies in plasma of 115 hospitalized COVID-19 patients. Statistical analysis was used to identify correlations between autoantibodies' levels and the severity of the disease. They propose that "anti-DNA autoantibodies could be used as a predictive biomarker for disease severity and specific clinical manifestations." Overall, the results of the research are in line with the emerging new hypothesis pointing to autoreactivity as a common phenomenon in COVID-19 patients.

The main limitation of this study is that autoantibodies against the lysate of erythrocytes, PS, and DNA react towards multiple antigens. Given previous literature, it is therefore not surprising to find these types of autoantibodies in COVID-19 patients. For instance, autoantibodies against b2GPI and prothrombin have been reported before in COVID-19 patients. These proteins are known to bind to negatively charged surfaces such as PS and lysate of erythrocytes. In this context, these findings provide new but incremental knowledge. Also, they do not address a key open question that is what autoantibody is doing what. Finally, the conclusion that there is a strong correlation between anti-DNA autoantibodies and disease severity would benefit from additional data points because of the large data spread shown in Figure 1B. While interesting, the outcome of the paper could be improved with the following suggestions.

Major comments

1) While mentioned several times throughout the text, it would be extremely informative to show, side-by-side, the levels of anti-RBCL, anti-PS, and anti-DNA found in COVID-19 patients alongside other infections and autoimmune diseases (such as SLE and APS) as well as provide a better description of their antigen specificity. In essence, what is the difference, if any, between anti-RBCL, anti-PS, and anti-DNA autoantibodies found in COVID-19 patients and other disease states? Information regarding autoantibody subtypes, in addition to IgG, is also desirable.

2) The circulating levels of IC reported in Figure 2 are surprisingly high for the control group. Also, their relationship with autoimmunity is unclear. Please explain.

3) COVID-19 patients were stratified into three groups: non-severe, severe patients who survived, and severe patients who died of COVID-19. However, no information is provided regarding their clinical history, specifically co-morbidities and medications, which is highly relevant for the search for autoantibodies.

4) For the anti-PS ELISA assay, after immobilization, PS-coated plates were washed 3 times with PBS 0.05% Tween-20 before blocking. Tween-20 is a nonionic detergent that is typically used for lysing cells as it destabilizes membrane bilayers. The use of Tween-20 is not recommended when investigating protein-lipid interactions. Please explain why the authors use Tween-20 in their ELISA assays and, if possible, provide evidence that Tween-20 does not interfere with the detection of anti-PS autoantibodies.

Minor comments

Ref 5 - Missing volume/ number/pages

Ref 11 - Update ref to the peer-reviewed article

Ref 13 - Missing volume/ number/pages

Ref 24 - Missing volume/ number/pages

Ref 28 - Missing volume/ number/pages

Referee #2 Review

Comments on Novelty/Model System for Author:

The study in my opinion is of clinical relevance. It describes for the first time that anti-DNA autoantibodies may play an important role in the pathogenesis of COVID-19 and could be developed as a predictive biomarker for disease severity and specific clinical manifestations.

Anti-DNA antibodies correlate with a marker of thrombosis, D-dimer. Anti-DNA antibodies also correlated with a parameter that determines the variation of erythrocyte volume, RDW, which has been associated with mortality risk in multiple diseases, including COVID-19.

These findings suggest that the most severe forms of disease observed in COVID-19 patients may be a result of the host response to infection, rather than a direct consequence of viral cytopathic effect. Autoantibodies, as part of the host response to infection, may contribute to this delayed pathogenesis through different mechanisms. Our work suggests that anti-DNA antibodies may have a role in different pathogenic processes, including cell injury and coagulation, constituting a possible mechanism contributing to pathogenesis in COVID-19 patients.

Remarks for Author:

In this study the Authors performed a retrospective study of 115 COVID-19 hospitalized patients with different severity to analyze the generation of autoantibodies to common antigens: a lysate of erythrocytes, the lipid phosphatidylserine (PS) and DNA. The statistical relation of autoantibody levels to death, disease severity and specific pathologies was determined using high-throughput data analysis of 118 clinical parameters for all patients.

They found that high levels of IgG autoantibodies against erythrocyte lysates were observed in a large percentage (up to 41%) of patients with COVID-19 compared to uninfected controls. Interestingly, they found that anti-DNA antibodies determined upon hospital admission were high in 16% of patients and correlated strongly with later development of severe disease, showing a positive predictive value of 89.5% and accounting for 22% of total severe cases. A statistically significant correlation was found between anti-DNA antibodies and markers of cell injury, coagulation, neutrophil levels and erythrocyte size.

They concluded that Anti-DNA autoantibodies may play an important role in the pathogenesis of COVID-19 and could be developed as a predictive biomarker for disease severity and specific clinical manifestations.

The study is of interest and of clinical relevance.

However, some issues deserve further details.

Patients characteristics: how the disease severity was defined? which criteria? chest CT? clinical?

Autoimmune antibodies: please specify if also other autoantibodies were searched for. To further improve the manuscript, in introduction or discussion, it would be interesting to discuss the results of the only study presently reporting the presence of autoantibodies in patients with COVID-19 possibly reflecting a pathogenetic role of immune dysregulation (COVID-19 and immunological dysregulation: can autoantibodies be useful? Clin Transl Sci. 2020 Sep 29;10.1111/cts.12908. doi: 10.1111/cts.12908).

A relevant issue is the detection of anti-DNA antibodies since it is well known that solid-phase assays are less specific than indirect immunofluorescence technique using *Critidia luciliae* (Diagnostic role of anti-dsDNA antibodies: do not forget autoimmune hepatitis. Nat Rev Rheumatol. 2021 Jan 18. doi: 10.1038/s41584-021-00573-7. Epub ahead of print). In the discussion, I suggest adding a paragraph discussing the potential limit of the used assay.

Referee #3 Review

Comments on Novelty/Model System for Author:

There are concerns regarding the anti-PS & anti-DNA assays (see details in the report to the authors).

Remarks for Author:

The paper reports a retrospective study on the presence of autoantibodies and circulating immune-complexes (CIC) in 115 COVID-19 hospitalized patients. The authors detected autoAbs against RBC-lysates, anti-PS & anti-DNA IgG and CIC by home-made assays. High levels of IgG autoAbs against RBC lysates were found in up to 41% of COVID-19 patients compared to uninfected controls. Anti-DNA

Abs were detected in 16% of patients and correlated with later development of severe disease,

markers of cell injury, coagulation, neutrophil levels & RBC size.

The main authors' conclusion is that autoimmune responses may play a role in the pathogenesis of COVID-19 and that anti-DNA autoAbs may be a biomarker for disease severity.

Main comments

1. As stated by the authors themselves, different infections (viral, bacterial & protozoan) may induce autoAbs against self-Ag. To support the authors' conclusions a control group of patients with another non-SARS-CoV-2 infection should be tested for the same autoAbs.
2. It seems that 20 uninfected normal (?) control samples were used for calculating the cutoff values of the home-made assays. A larger number is necessary and more information on age/sex etc distribution, how the normal values were calculated should be reported. For example, anti-phospholipid antibodies do not display Gaussian distribution and cutoff values calculated as mean+3SD are not acceptable.
3. The method for the detection of anti-PS Abs is not in line with the usual methods for the anti-phospholipid (aPL) antibodies. The described solid-phase detects Abs against PS and not against the PL-binding proteins (i.e. beta2GPI or prothrombin) which are actually the true autoimmune aPL. In this regard, any speculation on the significance of these autoAbs cannot be related to the anti-phospholipid syndrome or even to the thrombophilic state supported by autoimmune aPL. Further experiments to investigate the presence of true aPL (e.g. aCL, anti-beta2GPI or anti-PS/PT IgG/IgM) should be carried out.
4. Anti-DNA Abs: the described assay likely detects anti-single stranded DNA rather than anti-double stranded DNA. Control experiments with sera positive for anti-dsDNA should be carried out to characterize the specificity of the assay. This point is closely related to comment in point n.1. In fact, anti-ssDNA Abs are well known to be detectable in several infectious disease at variance of anti-dsDNA Abs found in SLE or SLE-like disease only.

Minor comment

1. If it is true that circulating autoAbs are playing a role in COVID-19 pathogenesis, do authors have data showing complement activation in their patients?
2. More details how the severity of the COVID-19 disease was defined should be reported.

2nd Review Round

Referee #1 Review

Remarks for Author:

The revised version of the manuscript has much improved. It now provides new important information enabling a more direct comparison between COVID-19 and other diseases, namely SLE and malaria, characterized by elevated titers of autoantibodies.

While I applaud the authors for the effort and amount of work, because of this new data, it appears that the presence and titers of anti-RBC, anti-PS, and anti-DNA antibodies are not a unique signature of COVID-19 patients; rather, they may represent a more general biomarker of autoreactivity. Furthermore, even though anti-DNA and anti-PS antibodies are (slightly) more elevated in severe COVID-19 patients than non-severe patients and controls, the differences between these groups are not striking. This observation is even more puzzling to interpret considering that, as written by the authors, the analyses were not adjusted for any other comorbidities or clinical history that might have impacted patient outcomes.

I have two additional comments:

- 1) The authors provide no direct evidence that these autoantibodies may play important roles in the disease pathogenesis, unlike in currently published work (DOI: 10.1111/jth.15455)
- 2) In this context, the choice of malaria is unclear. COVID-19 is an RNA virus. One would have expected to see comparative data between COVID-19 and a similar RNA virus, not a parasite.

August 2, 2021

Re: Life Science Alliance manuscript #LSA-2021-01180-T

Dr. Ana Rodriguez
Langone Medical Centre, NYU
Microbiology
Division of Parasitology
341 E 25th St
New York, NY 10010

Dear Dr. Rodriguez,

Thank you for submitting your manuscript entitled "Autoantibodies to DNA and to phosphatidylserine correlate with development of severe COVID-19" to Life Science Alliance. As indicated in the decision letter from our partner journal, I invite you to submit a revised manuscript addressing the following Reviewer comments.

-Address Reviewer 1's comment suggesting that this signature may be a more general biomarker of autoreactivity, as well as Additional Comment #2, via Discussion. Comment #1 should be addressed via Discussion, but no additional experimental data is required.

Thank you for this interesting contribution to Life Science Alliance. We are looking forward to receiving your revised manuscript.

Sincerely,

Eric Sawey, PhD
Executive Editor
Life Science Alliance
<http://www.lsa-journal.org>

- A letter addressing the reviewers' comments point by point.
- An editable version of the final text (.DOC or .DOCX) is needed for copyediting (no PDFs).
- High-resolution figure, supplementary figure and video files uploaded as individual files: See our detailed guidelines for preparing your production-ready images, <https://www.life-science-alliance.org/authors>
- Summary blurb (enter in submission system): A short text summarizing in a single sentence the study (max. 200 characters including spaces). This text is used in conjunction with the titles of papers, hence should be informative and complementary to the title and running title. It should describe the context and significance of the findings for a general readership; it should be written in the present tense and refer to the work in the third person. Author names should not be mentioned.

B. MANUSCRIPT ORGANIZATION AND FORMATTING:

Response to reviewers

Address Reviewer 1's comment suggesting that this signature may be a general biomarker of autoreactivity, as well as Additional Comment #2, via Discussion. Comment #1 should be addressed via Discussion, but no additional experimental data is required.

Reviewer 1 comment on signature being a general biomarker of autoreactivity:

We have observed that the levels of anti-RBC antibodies, which are actually a general biomarker for autoreactivity, do not correlate with or predict disease severity (Table III), indicating that the signature is not a general biomarker for autoreactivity. There is also evidence from other published work that while particular antibody specificities correlate with disease severity, levels of general autoreactivity do not (Wang et al., 2021). This is now mentioned in Discussion line 276.

Reviewer 1 comment on choice of malaria as comparison for autoreactivity:

The original request of the reviewer was to compare the autoreactivity levels of COVID-19 to other infectious disease and did not specify RNA virus. The choice of malaria was made on: 1) the previous knowledge that, similar to COVID-19, malaria is highly inflammatory and triggers a strong autoantibody response and 2) sample availability. The similarities between malaria and COVID-19 are highlighted on Discussion line 255.

We have modified the title to indicate more precisely that anti-DNA and anti-PS predict disease severity, since antibody determinations were made at the time of admittance to the hospital but severity was developed later in patients.

August 6, 2021

RE: Life Science Alliance Manuscript #LSA-2021-01180-TR

Dr. Ana Rodriguez
New York University School of Medicine
Microbiology
430 E 29th st.
New York, NY 10016

Dear Dr. Rodriguez,

Thank you for submitting your revised manuscript entitled "Autoimmune anti-DNA and anti-Phosphatidylserine antibodies predict development of severe COVID-19". We would be happy to publish your paper in Life Science Alliance pending final revisions necessary to meet our formatting guidelines.

- please consult our manuscript preparation guidelines <https://www.life-science-alliance.org/manuscript-prep> and make sure your manuscript sections are in the correct order
- please check that all Authors have been inserted in the Author Contribution section in the main manuscript text
- please use Arabic numbers when labeling the tables
- please add your main and table legends to the main manuscript text after the references section
- please revise the legends for figures so that the panels are introduced in order
- please add a conflict of interest statement to your main manuscript text
- please add callouts for Figures 1A-C, 2A-D, 3A-C to your main manuscript text

LSA now encourages authors to provide a 30-60 second video where the study is briefly explained. We will use these videos on social media to promote the published paper and the presenting author. Corresponding or first-authors are welcome to submit the video. Please submit only one video per manuscript. The video can be emailed to contact@life-science-alliance.org

A. FINAL FILES:

B. MANUSCRIPT ORGANIZATION AND FORMATTING:

Sincerely,

Eric Sawey, PhD
Executive Editor

August 18, 2021

RE: Life Science Alliance Manuscript #LSA-2021-01180-TRR

Dr. Ana Rodriguez
New York University School of Medicine
Microbiology
430 E 29th st.
New York, NY 10016

Dear Dr. Rodriguez,

Thank you for submitting your Research Article entitled "Autoimmune anti-DNA and anti-Phosphatidylserine antibodies predict development of severe COVID-19". It is a pleasure to let you know that your manuscript is now accepted for publication in Life Science Alliance. Congratulations on this interesting work.

DISTRIBUTION OF MATERIALS:

Again, congratulations on a very nice paper. I hope you found the review process to be constructive and are pleased with how the manuscript was handled editorially. We look forward to future exciting submissions from your lab.

Sincerely,
